# Management of Autism Spectrum Disorder in Italian Units of Child and Adolescent Mental Health: Diagnostic and Referral Pathways

**DOI:** 10.3390/brainsci12020263

**Published:** 2022-02-14

**Authors:** Marta Borgi, Flavia Chiarotti, Gianfranco Aresu, Filippo Gitti, Elisa Fazzi, Angiolo Pierini, Teresa Sebastiani, Marco Marcelli, Renato Scifo, Paolo Stagi, Aldina Venerosi

**Affiliations:** 1Center for Behavioral Sciences and Mental Health, Istituto Superiore di Sanità, Viale Regina Elena 299, 00161 Rome, Italy; marta.borgi@iss.it (M.B.); flavia.chiarotti@iss.it (F.C.); 2Child and Adolescent Neuropsychiatric Unit, Azienda Regionale per la Tutela della Salute, S.S. 200 (Sassari-Sorso) dell’Anglona, Località San Camillo, 07100 Sassari, Italy; gianfrancoaresu@tiscali.it; 3Child Neurology and Psychiatry Unit, ASST Spedali Civili of Brescia, Piazzale Spedali Civili 1, 25123 Brescia, Italy; filippo.gitti@asst-bergamoest.it (F.G.); elisa.fazzi@unibs.it (E.F.); 4Department of Clinical and Experimental Sciences, University of Brescia, 25121 Brescia, Italy; 5Department of Child Neuropsychiatry, USL Umbria 1, Via della Pallotta 42, 06126 Perugia, Italy; angiolo.pierini@uslumbria1.it; 6Child and Adolescent Neuropsychiatric Unit, Azienda Sanitaria Locale Viterbo, Via Enrico Fermi 15, 01100 Viterbo, Italy; teresasebastiani77@gmail.com (T.S.); sdif.segr@asl.vt.it (M.M.); 7Center for Autism Spectrum Disorders, Child Psychiatry Unit, Provincial Health Service of Catania (ASP CT), Via T.M. Manzella, 95125 Catania, Italy; renato.scifo@aspct.it; 8Child and Adolescent Mental Health Service AUSL, Via S. Giovanni del cantone 23, 41121 Modena, Italy; paolo.stagi@uslcentro.toscana.it

**Keywords:** autism spectrum disorder, child-adolescent mental health services, diagnosis, systems of care, Italy, individual health record

## Abstract

Overall, the present pilot study provides detailed information on clinical management for Autism Spectrum Disorder (ASD) referral and diagnosis processes that are mandatory for child and adolescent mental health management. The analysis of ASD management, even if carried out on a selected sample of Child and Adolescent Mental Health (CAMH) units, represents a good approximation of how, in Italian outpatient settings, children and adolescents with ASD are recognised and eventually diagnosed. One of the aims of the study was to verify the adherence of Italian CAMH units to international recommendations for ASD referral and diagnosis and whether these processes can be traced using individual chart reports. Overall, the analysis evidenced that Italian CAMH units adopt an acceptable standard for ASD diagnosis, although the reporting of the ASD managing process in the individual chart is not always accurate. Furthermore, data collected suggest some improvements that CAMH units should implement to fill the gap with international recommendations, namely, establishing a multidisciplinary team for diagnosis, improving the assessment of physical and mental conditions by the use of standardised tools, implementing a specific assessment for challenging behaviours that could allow timely and specific planning of intervention.

## 1. Introduction

Autism spectrum disorder (ASD) is a neurodevelopmental disorder characterized by early-onset difficulties in social interaction, communication and stereotyped repetitive behaviours and interests. ASD affects approximately 1% of the population in high-income countries [1,2,3].

The disorder is lifelong, and people on the spectrum are reported to have elevated mortality risk [4], lower educational level, reduced quality of life and higher frequency of comorbid disorders, e.g., depression and anxiety [5,6]. Early intervention is essential to help with core features, behaviours and problems commonly associated with the condition [7,8,9]. However, delays in diagnosis are common [10,11] and specific interventions based on the best available evidence are lacking [7,12,13,14]. This causes significant suffering to individuals and families, leading to substantial service costs across several systems [15]. Children and young people with ASD frequently have adverse experiences in accessing health care and other services; accessibility and quality of care for ASD are indeed heterogeneous and frequently inadequate [16].

The present pilot study aimed to understand the current care pathway for children and adolescents with autism in the age range 1–17 yrs in a sample of Child and Adolescent Mental Health (CAMH) units in Italy. In particular, we investigated the ASD referral and diagnostic processes in a selected sample of Italian CAMH units in order to enhance our knowledge of the degree to which health professionals rely on national and international guidelines in their clinical practices. Several scholars and guidelines [8,17] have underlined the importance of early identification and referral for diagnostic evaluation and intervention services. Indeed, early diagnosis—paired with a timely intervention—appears to have a positive impact on life trajectory, even if its efficacy in reducing autism severity is low [18,19]. Hence, many countries, including Italy, are putting into place different actions to improve neurodevelopmental surveillance and increase knowledge and awareness about ASD symptoms and their early manifestation, mainly through training programs for health professionals and teachers and by promoting the use of standardised checklists.

In Italy, the institutional agreement between the Ministry of Health and the regions (hereinafter referred to as the Italian ASD Action Plan, IAAP; [20,21]) as well as the Italian law on ASD (Law n. 134/2015) identified strategic priorities for ASD health care, with the final aim of promoting harmonized protocols for ASD diagnosis and evidence-based intervention approaches, according to recommendations defined at the national level [22] and/or those that are of methodological reference for Italy at the international level [14]. In particular, these documents state the fundamental principles for ASD management: (i) capillary early diagnostic processes; (ii) comprehensive, easily accessible and widespread health service networks throughout the territory; (iii) multiprofessionalism and interdisciplinary synergy for the intervention that services must be able to offer; (iv) strong integration of the health, social, school and educational dimensions and (v) continuity of taking charge for the entire life of the person, with the progressive adaptation of interventions and the organization of living spaces. For the implementation of IAAP, CAMH units are strategic because they have a capillary distribution in the territory, each one serving a population of about 100,000 residents, of whom—on average in 2017—about 16% are under 18 years of age (this percentage strongly varying across time and geographical areas). Hence, CAMH units are supposed to reach, with good approximation, a great part of the resident population. However, a recent Italian nationwide survey—aimed to analyse structural capacity (i.e., allocation and provision of qualified human resources) and ASD services provided to ASD patients, addressed to Italian CAMH units—showed a great geographical heterogeneity of the services available and an insufficient ability to provide the interventions recommended at national and international levels [23]. The survey has indeed shown that about the 72% of CAMH units collaborate with paediatricians but only 59% with the school, although this is compulsory in the legislation. This can be due to an investment shortage for the area of the neuropsychiatry of childhood and adolescence in Italy. This interpretation is supported by the quantitative analysis of human and technological resources, which appear to be inadequate in most of the CAMH units [23]. In addition, the analysis of the specific legislation published at the health district and regional levels showed a strong fragmentation of the regulator’s actions and a clear heterogeneity between the Italian regions. Among the critical issues that the survey detected, there was a poor attitude toward the digitization of clinical data (present in 60% of the CAMHs in Italy, ranging from 41% in the islands to 82% in the northern regions). The poor development of computerized archives related to CAMH activities causes several gaps in knowledge affecting ASD management. Improvement of the knowledge of the ASD nosography and monitoring of the ASD health policies would allow verifying accessibility and unmet needs of children and adolescents with ASD. The capillary organization of the Italian Health System (with the CAMH units widely distributed in the territory) facilitates the systematic and digital collection of clinical data in representative samples of the Italian ASD population and the implementation of epidemiological studies.

Crucial to the adoption of computerized clinical archives is the implementation of a structured electronic protocol able to collect the set of individual information currently gathered during the clinical visit. In the present study, we defined a structured protocol composed of a set of variables describing the expected managing processes for children and adolescents diagnosed with ASD. By a retrospective analysis conducted by a chart review of records from a selected sample of CAMH units, we assessed if the information contained in the current chart records matched the structured protocol. Our final aim was to evaluate the quality and the completeness of the CAMH medical records with respect to the adoption of ASD management procedures, following national and international guidelines, and their change in time.

## 2. Materials and Methods

### 2.1. Participants and Procedure

The inclusion criteria for the selection of the CAMH units were: (i) the presence of a formal pathway for managing ASD for more than five years (information on CAMHs was collected in a previous study, see most of the CAMH units [23]); (ii) the availability of a specialist responsible for the ASD pathway from the CAMH unit who could participate in consensus meetings with the researchers leading the project and (iii) the provision of data collected in clinical records for a selected sample of children in at least two age groups (see below the criteria for the selection of the clinical population). The CAMH units were also selected based on their geographical distribution (see Table 1).

CAMH units provided the individual patient data collected in clinical records using a structured Electronic Collection Protocol (here denominated ECP) that was made available online. The ECP was the result of a consensus process shared in the working group composed of researchers of the Istituto Superiore di Sanità (project leader) and the specialists in charge of the ASD pathways in the CAMHs. Three meetings were held and the discussion was aimed to define the group of selected variables able to describe the main steps of the referral and diagnosis processes, taking into account the most recent international and national (Italy) evidence-based recommendation available in the year 2019 [14,22].

CAMH units were asked to provide individual medical records, following the list of variables in the ECP. Data collected in the participating CAMH units refer to children and adolescents belonging to three groups based on age reached on 31 December 2016 (age in completed years): subjects born between 1 January 2011 and 31 December 2014 (1–4 yrs age group), subjects born between 1 January 2004 and 31 December 2007 (8–11 yrs age group) and subjects born between 1 January 1998 and 31 December 2001 (14–17 yrs age group). Specifically, one CAMH unit in the Centre provides services only to children and adolescents up to 14 years of age, while one CAMH unit in the Islands area could not retrieve individual medical records for adolescents in the 14–17 yrs age group. Each individual clinical record was selected by a chronological criterion (children entering the study are extracted by the date of creation of their clinical record at CAMH unit, separately for each group of age). We considered it sufficient for our analysis to reach a number of children and adolescents equal to or greater than the 20% of ASD subjects afferent to the CAMH units.

### 2.2. Data Collection

Children’s data were collected retrospectively from their medical records and were analysed anonymously. A Moodle^®^ platform was used to collect data online. Telephone support was provided to the respondents.

### 2.3. Measures

The structure of the ECP was based on a conceptual framework containing relevant dimensions derived from national and international recommendations for ASD referral and diagnosis. More particularly, the ECP included a list of variables structured in the following four sections: (i) socio-demographic information relative to children (birth year and sex) and parents (birth year, nationality, education, employment and marital status); (ii) referral (referrer and motivation/concern for referral); (iii) diagnosis and clinical characterization (diagnostic category based on ICD-10, age at diagnosis, presence of mental retardation and challenging behaviours, presence of coexisting mental and physical health conditions, laboratory tests, composition of the team involved in diagnosis and use of standardized tools during the diagnostic process) and (iv) standard of the diagnosis process (presence of written clinical assessment report, wait time from referral/initial concern to the first visit, wait time from the first visit to diagnosis and wait time from diagnosis to the beginning of the intervention).

The ECP referred to the ICD-10 codes for ASD diagnosis and presence of mental retardation assessment as well as for the assessment of coexisting mental and physical health conditions. As for challenging behaviours, the ECP referred to the American Psychological Association (APA) definition: “Behavior that is dangerous or that interferes in participation in preschool, educational, or adult services and often necessitates the design and use of special interventions” (https://dictionary.apa.org/challenging-behavior, last visited on 23 November 2021).

In the protocol, participants were asked to report the use of the following standardized tools: (i) diagnosis: Autism Diagnostic Observation Schedule—Generic (ADOS-G), Autism Diagnostic Observation Schedule—Generic, Second Edition (ADOS-2), Childhood Autism Rating Scale (CARS) and Checklist for Autism Spectrum Disorder (CASD); (ii) cognitive functioning: Griffiths Scales of Child Development (GRIFFITHS), Bayley Scales of Infant Development (BAYLEY), Leiter International Performance Scale—Revised (LEITER), Wechsler Preschool and Primary Scale of Intelligence (WPPSI-III) and Wechsler Intelligence Scale for Children (WISC-III); (iii) adaptive skills: Vineland Adaptive Behavior Scale (VABS) and (iv) definition of the psychoeducational profile: Psycho-Educational Profile (PEP), Psycho-Educational Profile, 3rd Edition (PEP-3) and Early Start Denver Model (ESDM). For each category (diagnosis, cognitive functioning, adaptive skills and definition of the psychoeducational profile), we computed the frequency of subjects undergoing the evaluation by at least one of the tools listed in the category. Autism Diagnostic Interview—Revised (ADI-R) was not included in the diagnosis category, as this tool resulted to be very rarely used in the participating CAMH units (n = 20), never alone but always in combination with ADOS or CARS.

With respect to professionals involved in the diagnosis process, the recent SIGN evidence-based guideline [14] stated “diagnostic assessment, alongside a profile of the individual’s strengths and weaknesses, carried out by a multidisciplinary team which has the skills and experience to undertake the assessments, should be considered as the optimum approach for individuals suspected of having ASD”. Furthermore, the quality standard by NICE states, in QS15, that the team should include paediatricians and/or child and adolescent psychiatrists, speech and language therapists and clinical and/or educational psychologists. According to this standard, we classified team composition as optimal, suboptimal, sufficient and poor with respect to the degree of multidisciplinarity of the teams involved in the process. In particular, we attributed the following weights to the professionals involved in the teams, based on their academic profiles and degrees of autonomy in performing a comprehensive diagnosis of all aspects (ASD, cognitive functioning, language and adaptive skills) using the specific tests: child and adolescent psychiatrist (weight = 3), psychologist (weight = 2), speech therapist (weight = 1.5) and neurodevelopmental disorders therapist and educator (weight = 1). Teams achieving scores more than 5 were classified as optimal, between 3.5 and 5 as suboptimal, equal to 3 as sufficient and less than 3 as poor.

### 2.4. Data Analysis

Categorical variables were described using absolute and percent frequencies, while quantitative variables were summarized by medians and ranges. The Fisher’s exact probability test was used to compare the distribution of categorical variables between CAMH units within age groups and between age groups within CAMH units. Comparisons among units with respect to the quantitative variables were performed using the Kruskal–Wallis test to take into account possible non-normality of data and/or heteroscedasticity among subgroups. Specifically, the Kruskal–Wallis test was performed to compare the four or six CAMH units within each age group and age groups within each CAMH unit. Analyses were performed using STATA (Stata Statistical Software, Release 16.0. College Station, TX, USA: Stata Corporation).

## 3. Results

### 3.1. Sample Characteristics

Six CAMHs participated in the survey, providing information on 634 children and adolescents with ASD who met the inclusion criteria (see Methods—Participants and Procedure). Table 2 shows the distribution of the subjects by age group and sex in the six participating CAMH units.

The overall sex ratio (M:F) was 4.4, ranging from 3.2 to 7.8 in the different CAMH units. Specifically, the sex ratio was 4.6 (range 2.5–12) in the 14-17 yrs age group, 6.0 (range 3.4–11.3) in the 8–11 yrs age group and 3.4 (range 2.1–6.9) in the 1–4 yrs group.

The ICD-10 classification of subjects is reported in Appendix A. Overall, diagnostic categories reported were: F84.0 in about 50–60% of the subjects, depending on the subgroup of CAMHs, and F84.9 in more than 20% of subjects, while subjects within the categories F84, F84.1, F84.5 and F84.8 were all less than 15% of the sample (Appendix A).

Almost half (about 45–50%) of the subjects had mental retardation (F70–F79; see Appendix A). Within the subjects with information on both mental retardation and challenging behaviours, more than 34% had neither mental retardation nor challenging behaviours, while 26% had both (Appendix A). Challenging behaviours were detected in about 40% of subjects. Interestingly, this percentage was very similar in the three age groups (CAMH A-D: 40.4%, 43.0% and 32.6% for 14–17 yrs, 8–11 yrs and 1–4 yrs, respectively; CAMH A-F: 42.0% and 41.5% for 8–11 yrs and 1–4 yrs, respectively; Appendix A).

Information on coexisting mental health conditions was not available for 109 (17.2%) children and adolescents (14–17 yrs: n = 8; 8–11 yrs: n = 47; 1–4 yrs: n = 54). Of those for whom this information was available (n = 525), 26.5% (n = 139) had mental health conditions.

As for physical health conditions, information was not available for 121 (19.1%) children and adolescents (14–17 yrs: n = 10; 8–11 yrs: n = 50; 1–4 yrs: n = 61). Of those for whom this information was available (n = 513), 17.0% (n = 87) showed association with physical health conditions.

For each CAMH unit or group of units, we observed a significant difference among age groups for most of the analysed variables concerning referral, ASD diagnosis and clinical characterization and standard of the diagnostic process (see Appendix A for statistical significance levels). For this reason, data were analysed, and results are reported, separately in the three age groups, in the following.

### 3.2. Referral

Information on the referrer was not available for 18 (2.8%) children and adolescents (14–17 yrs: n = 2; 8–11 yrs: n = 7; 1–4 yrs: n = 9). Table 3 shows the distribution of subjects according to the referrer in the different CAMH units and age groups. We observed a large variability across age groups and CAMH units in the frequencies of referrers. School, paediatricians and family were the main referrers, either when considering the four CAMH units reporting information for all age groups or the six CAMH units for subjects aged 11 years or less. In particular, in the 1–4 yrs age group, we can observe a decrease in the frequency of school referral and a parallel increase in the frequency of paediatrician referral.

Information about the neurodevelopmental concerns for referral was not available for 38 (6.0%) children and adolescents (14–17 yrs: n = 23; 8–11 yrs: n = 14; 1–4 yrs: n = 1). Table 4 describes in detail the single and multiple concerns for referral reported in the individual clinical chart. Data demonstrated a high heterogeneity among CAMH units and did not evidence a specific trend among the age groups. However, the ASD, Language, Social and Soc + Lang (Social and Language impairments) categories were the most frequently detected in all age groups. In particular, Social and/or Language accounted for about 50% of all concerns for referral. Interestingly, a greater frequency of ASD concern for referral was shown in the 1–4 yrs group, either when considering the four CAMH units reporting information for all the age groups or the six CAMH units for subjects aged 11 years or less.

### 3.3. ASD Diagnosis and Clinical Characterization

Information relative to age at diagnosis was not available for 19 (3.0%) children and adolescents (14–17 yrs: n = 3; 8–11 yrs: n = 7; 1–4 yrs: n = 9). The data showed that age at diagnosis decreased between the 14–17 and 8–11 yrs age groups for most of the CAMH units. These age groups cannot be compared with the 1–4 yrs age group: indeed, in the latter group, only children diagnosed under the age of five years were included, thus introducing a ceiling effect on the estimation of the median age at diagnosis (Table 5).

As for the professionals involved in the diagnosis and clinical characterization, information was not available for 16 (2.5%) children and adolescents (14–17 yrs: n = 1; 8–11 yrs: n = 6; 1–4 yrs: n = 9).

Table 6 shows the distribution of subjects according to team composition in the different CAMH units and age groups. In most CAMH units and age groups, more than 80% of diagnoses were provided by teams of optimal/suboptimal composition, except for CAMH B in the 1–4 yrs age group, where this percentage was about 59%. The poor team composition was shown only in about 20% of diagnoses in CAMH E. As none of the ASD diagnoses and clinical characterizations were provided by teams classified as good, this class was not reported in Table 6.

CAMH units were asked to report the standardized tools used during the diagnostic process. Information was not available for 61 (9.6%) children and adolescents (14–17 yrs: n = 22; 8–11 yrs: n = 14; 1–4 yrs: n = 25). Table 7 shows the frequency of subjects assessed with standardized tools for: (i) diagnosis; (ii) cognitive functions; (iii) adaptive skills and (iv) definition of the psychoeducational profile. Overall, data evidence a high heterogeneity among CAMH units. Data on the frequency of subjects diagnosed by at least one standardized tool among ADOS-G, ADOS-2, CARS and CASD showed increased frequency across the age groups. The highest frequency was shown in the 1–4 yrs age group, both in CAMH units reporting information for all the age groups and in the six CAMH units for subjects aged 11 years or less. As for cognitive assessment, less use of standardized tools was observed in the 1–4 yrs age group as compared to the 8–11 and 14–17 yrs age groups that appear more similar. The frequency of the use of VABS showed high heterogeneity among CAMH units, both overall and as a trend across age groups. Finally, standardized tools for the definition of psychoeducational profiles were used only in some CAMH units (B, E and F). In these, data showed a higher frequency of use in the 1–4 yrs age group.

With respect to other clinical characterization by laboratory tests, information was not available for 162 (25.6%) children and adolescents (14–17 yrs: n = 38; 8–11 yrs: n = 52; 1–4 yrs: n = 72). An overall heterogeneity was observed among CAMH units within age groups and among age groups within CAMH units. Absolute and percent frequencies of subjects undergoing the different laboratory tests by CAMH unit and age groups are reported in Appendix A. The laboratory test most frequently performed was EEG (about 60% in 1–4 yrs children and more than 70% in subjects aged 8 yrs or more). Auxological, auditory, genetic and magnetic resonance assessments were performed in a percentage of subjects, ranging from 30 to 55%.

### 3.4. Standard of the Diagnosis Process

Written clinical assessment reports to be shared with patients/caregivers are standards for the diagnosis process. This information was not available for 109 (17.2%) children and adolescents (14–17 yrs: n = 23; 8–11 yrs: n = 52; 1–4 yrs: n = 34). When information was available, the frequency of subjects that had received a written report was overall near 100% in the 1–4 yrs age group and above 80% for the other age groups (Appendix A).

The timing of the diagnosis process was described by three variables concerning the wait time: (1) from referral/initial concern to first visit, (2) from first visit to diagnosis and (3) from diagnosis to start of intervention. We set the following standard: We categorized Variable 1 as “<= 90 days” and “> 90 days”. For Variables 2 and 3, we set the cut-off at “<=90 days” and “<=180 days”, respectively. Information for Variables 1 and 3 was missing for many subjects across all CAMH units (227 or 35.8% and 220 or 34.7% for variables 1 and 3, respectively). Information for Variable 2 was missing for 85 subjects (13.4%).

As for the wait time between referral/initial concern and first visit (Variable 1), available individual chart data evidenced a good standard for all the CAMH units analysed, with the exception of the CAMH unit F, which showed only 50% of the subjects in the 8–11 yrs age group with waiting times <=90 days (Appendix A). Analysis of the wait time for obtaining a diagnosis (Variable 2) evidenced that the percentage of subjects receiving a diagnosis within 90 days from the first visit was higher than 50% in CAMH units B, E and F and lower than 50% in CAMH units A, C and D. Individual chart data also evidenced overall poor compliance with the “<=180 days” cut-off for wait time between diagnosis and intervention (Variable 3), with a percentage of subjects within the cut-off of 180 days always under 50% in the 14–17 and 8–11 yrs age groups. In the 1–4 yrs age group, this percentage was higher, especially in CAMH units B (97.4%) and E (72.7%).

## 4. Discussion

The aim of the study was to understand if Italian CAMH units adhere to international recommendations for ASD referral and diagnosis management and if this behaviour could be traced by means of individual chart reports. The analysis of ASD management was conducted in a selected sample of CAMH units with good expertise in ASD care for children and adolescents. Overall, the analysis evidenced that Italian CAMH units adopted an acceptable standard for ASD diagnosis, even if the reporting of the ASD managing process in individual charts was not always accurate.

The subjects included in the present study showed characteristics comparable with those reported in the international literature. In particular, the overall male-to-female ratio observed in our sample was 4.4:1, a value very similar to that reported in recent ASD surveillance in Italy [2] and other countries (e.g., USA reported by the CDC, [24]). However, the high variability of the sex ratio observed in our Italian sample suggests the presence of a gender bias in ASD diagnosis, as reported in the literature [25].

Moreover, the presence of intellectual disability (47.2% of subjects) fell in the range between the percentage reported by the CDC (35.2%, [24]) and that reported by the Italian prevalence study (66%, [2]). Associated mental and physical conditions were reported for about 26% and 18% of the subjects, respectively. These percentages appear well below those reported in the literature (see [26,27] for reporting percentages of subjects with specific mental and/or physical conditions associated with ASD) and confirm the need to establish and improve a comprehensive global health assessment of coexisting physical and mental disorders during diagnosis. Moreover, although data were available for the majority of subjects (proportion of subjects with missing data was below 20%), high variability among CAMH units in the incidence of mental and physical conditions was reported. It is important to consider that our study was not a prevalence study; individual data were collected for different age groups by a retrospective analysis of diagnosis reports. Nevertheless, the sex ratio and prevalence of intellectual disability in our sample are still comparable with international prevalence reports, suggesting the effectiveness of Italian CAMH units in detecting ASD in the general population and in acting as surveillance stations to monitor ASD epidemiology.

In the present study, we included subjects born in three different calendar year periods to verify if management processes have changed over time as an effect of cultural changes and/or implementation of specific ASD health policy strategies. As for referral, data showed an increasingly marginal role of the school in the referral process and a corresponding increase in the frequency of referral by paediatricians for early evaluation of neurodevelopmental disorders. This trend is particularly evident in the group of children aged 1–4 yrs, that is, the group of children that received diagnoses more recently. This consideration is also supported by data showing an increase over time in the use of M-CHAT by paediatricians (higher frequency in the 1–4 yrs age group). Both an increased awareness about autism “red flags” in the child’s living environments and an enhanced inclination of paediatricians to neurobehavioural development surveillance, including the use of ASD screening tools (e.g., M-CHAT), could account for this result. It is important to note that the role of paediatricians in the referral process is particularly evident in the CAMH units involved in specific programs aimed at establishing the role of primary care for the detection of ASD. In Italy, various local experiences aimed at connecting the paediatricians to CAMH units to facilitate the timing of ASD recognition have been launched in the last ten years. These programs have tried to promote the use of checklists and/or clinical evaluation of early signs of ASD in the primary care paediatric setting. In line with the international recommendations on ASD surveillance [8], these programs were also included in the IAAP [20,21] as good practices to be integrated into regional health system regulations. Heterogeneity shown by our data concerning both the main source of referral and concern for referral might be the result of a different implementation of the IAAP at the regional/district level and suggests the need for a clear supraregional mandate that helps local services to converge on these practices. School and family remain two important sources able to inform on early warning signs of autism. However, the involvement of an intermediate health referrer (such as the paediatrician) who conveys the information to services in charge of the diagnostic assessment might be more effective and efficient from the point of view of public health. One further important point that should be investigated is the presence of disparities in access to essential services. Recently, Wallis and collaborators [28] reported disparities in ASD diagnosis and intervention based on children’s sex, language, socio-economic status and race. This and other reports call for future research aimed at clarifying the role of social determinants in affecting variability in ASD-related utilization of health services.

In most of the CAMH units, information collected at age at diagnosis showed an earlier ASD diagnosis in subjects in the 8–11 yrs group when compared to older children (14–17 years old), with a difference spanning from a minimum of 3 months to a maximum of 12 months. The recruitment criterion did not allow us to compare the 1–4 yrs age group with the other age groups since subjects were enrolled in the former group only if diagnosed within the fourth year of age (ceiling effect). A follow-up study would be useful to verify if the trend shown is stable over time.

The diagnosis of ASD is performed by differentiating between two or more conditions that share similar signs or symptoms. ICD-11 and/or DSM-5 define and classify mental disorders in order to improve diagnoses, treatment and research, though many tools have been developed and validated to operate a more structured assessment. As for ASD diagnosis, ADOS as well as CARS are considered gold standards. Results from the analysis of the individual chart reports showed increased use over time of these instruments for ASD diagnosis in all the CAMH units considered. More surprisingly, a decrease in the assessment of cognitive levels by means of standardized tools was shown. This was also true for adaptive skills, although characterized by a high heterogeneity among CAMH units. However, in some units, an increasing trend to use structured tools for the definition of the psychoeducational profile was observed. This was specifically linked to the introduction of structured interventions, especially TEACCH and ESDM interventions, that target areas of strength and/or weakness related to different functional domains. Importantly, in our sample, the presence of coexisting mental conditions did not appear to be assessed by the use of standardised tools.

With respect to biomedical investigations offered to ASD patients, an overall heterogeneity was observed among CAMH units, as well as a quite large number of missing data (25%). Such heterogeneity and lack of information might be due to the presence/absence of an established connection of the CAMH unit with specialized services (i.e., highly specialized hospitals). Genetic assessment, EEG and MRI were the tests most frequently offered across CAMH units. As for differences among age groups, lower percentages of subjects undergoing genetic/karyotype investigations, EEG and MRI were observed in the 1–4 yrs age group than in 14–17 and 8–11 yrs age groups. This trend was very similar in all CAMH units and could be interpreted as a “cultural effect”, resulting from a different inclination to perform biomedical investigation depending on updates in the most recent research. However, it is important to consider that biomedical investigations as those here described are time- and cost-effective, and an accurate assessment of the predictive value of each new analysis has to be performed to preserve public health efficacy and efficiency. Similar considerations were reported in the international debate about potential ASD biomarker candidates [29]. As research progresses, genetic testing may contribute to identifying effective interventions related to specific aetiologies. ASD can be associated with a wide range of underlying conditions including genetic abnormalities. As an example, a chromosomal microarray analysis should be performed in the case of suspected genetic syndromes (e.g., coexisting intellectual disability, three-generation family history or presence of dysmorphic features). In some cases, genetic counselling should be offered in order to verify the presence of specific mutations potentially predictive of neurometabolic and/or immunological alterations [30,31]. Moreover, since young children with ASD differ from controls in several brain area volumes, neuroimaging assessments (as MRIs) appear to be promising diagnostic in-depth analyses. However, biomedical techniques such as MRI and EEG are not currently recommended as routine diagnostic practices because of the presence of conflicting results about their utility as biomarkers [14,32,33].

Recently, the National Institute for Health and Care Excellence (NICE) provided a quality standard for organization of ASD services (QS15; [13]). QS15 includes eight statements of quality and relative quantitative indicators. In particular, Statements 1 and 2 deal with standards for diagnosis. Statement 1, which focuses on the quality of the diagnostic process, requires that people with suspected ASD are referred to an autism team and their assessment is started within 3 months (90 days) of their referral. Our study examined waiting times for diagnostic assessment of ASD (wait time between referral/first concern and first visit). For the vast majority of subjects (about 90%), compliance with the quality standard (Statement 1) was shown, supporting a fairly good quality level of the Italian CAMH units for this indicator. As for Statement 2, NICE Q15 requires that people having a diagnostic assessment for autism are also assessed for coexisting physical health conditions and mental health problems. As previously described, the availability of individual data relative to mental and physical condition assessment was acceptable for these variables (subjects with missing data were 17.2% and 19.1%, respectively): Italian CAMHs showed a stable and sufficient inclination to conduct a global assessment, even in the presence of large variability among CAMH units.

The composition of the teams in the participating units appears to adhere to that defined as standard by NICE QS15 (a team should include: paediatricians and/or child and adolescent psychiatrists, speech and language therapists and clinical and/or educational psychologists). The analysis of the composition of the team involved in the ASD diagnostic process in our sample revealed that more than 85% of the individual charts examined in the Italian CAMH units reported a diagnostic assessment performed by an optimal and suboptimal team, as recommended by NICE QS15. Only a few subjects (n = 85) received a diagnostic assessment by a team of sufficient and/or poor level. However, very high heterogeneity was observed among CAMH units and within each CAMH unit among age groups. This observation suggests that the availability of professional staff can vary over time, probably due to the shortage of investment in the sector and/or to staff retirements. However, data were characterized by high variability among CAMH units and across age groups, suggesting that the composition of the autism team is affected by capacity factors.

Overall, this study shows a very high heterogeneity between CAMH units relative to both referral and diagnosis. As previously suggested [23], the regional organization of the Italian health system could have affected the harmonisation of clinical practices, probably affecting the quality of the services provided in some geographical areas. The recent delivery of the IAAP, as well as the presence of a specific law that advocates the needs of children, adolescents and adults with ASD, did not appear to have sufficiently guaranteed the availability of a defined referral pathway or the presence of a multidisciplinary team for ASD diagnosis and clinical characterisation.

In conclusion, data collected suggest some actions that CAMH units could implement to fill the gap with international recommendations: namely, to establish a multidisciplinary team for diagnosis; to improve the assessment of physical and mental conditions by the use of standardised tools and taking into account how ASD manifests differently in girls and boys and to implement a specific assessment for challenging behaviours that could allow timely planning of a targeting intervention. Furthermore, attention to the systematic and digital collection of sociodemographic data of users and their families should be improved, since assessing social and nonsocial determinants of health—especially in the case of mental disorders—is of paramount relevance.

## 5. Limitations

The main limitation of the present study was the low number of CAMH units included. However, the selected units represented the main Italian geographical areas. Regional administration of health in Italy could affect ASD management, thus collecting data from different geographical areas added information useful to interpret data variability. The second limitation was that the study was not a rigorous population-based study. However, one of our aims was to verify if the data collected in the individual charts included variables crucial to describe ASD diagnosis management. Data showed that Italian CAMH units are able to collect sufficient data in the individual charts, although there is some margin for improvement. Our study should be considered a pilot study, possibly acting as a motivation/drive to establish a harmonized digital system for ASD data collection on a selected number of indicators. Finally, it would be of particular interest to examine our data while also taking into account different standards of diagnostic assessment for ASD that are developed at international levels [34,35,36].

## 6. Conclusions

The set of variables used in the present study appears to be a useful starting point to build a database aimed at assessing the effectiveness of health policy interventions (e.g., age at the first diagnosis as an indicator of the efficacy of programs for early diagnosis involving paediatricians). Furthermore, the implementation of a database that includes accurate sociodemographic variables could be of paramount importance to verify the contribution of social determinants on the rate of referral and/or diagnosis.

## Figures and Tables

**Table 1 brainsci-12-00263-t001:** Geographical distribution of the selected CAMH units.

Area	ID CAMH Unit
NORTH (Lombardia, Emila Romagna)	CAMH unit A, CAMH unit D
CENTRE (Umbria, Lazio)	CAMH unit C, CAMH unit E
SOUTH/ISLAND (Sicilia, Sardegna)	CAMH unit B, CAMH unit F

CAMH: Child and Adolescent Mental Health.

**Table 2 brainsci-12-00263-t002:** Distribution of subjects by CAMH unit, age group and sex.

	Age Groups												
	14–17 yrs			8–11 yrs			1–4 yrs			All Ages			
CAMH	M	F	Tot	M	F	Tot	M	F	Tot	M	F	Tot	M:F
A	37	7	44	34	6	40	29	14	43	100	27	127	3.7
B	34	5	39	31	9	40	31	10	41	96	24	120	4.0
C	12	1	13	34	3	37	28	6	34	74	10	84	7.4
D	28	11	39	32	8	40	36	11	47	96	30	126	3.2
E	-	-	-	38	4	42	48	7	55	86	11	97	7.8
F	-	-	-	36	4	40	29	11	40	65	15	80	4.3
Tot	111	24	135	205	34	239	201	59	260	517	117	634	4.4

**Table 3 brainsci-12-00263-t003:** Frequency of subjects by referrer, CAMH unit and age group.

Centre	Age Group	CAP		Hospital		School		Paediatrician		Family		More than One *		Other		Total	
		n	%	n	%	n	%	n	%	n	%	n	%	n	%	n	%
**A**	**14–17 yrs**	1	2.3	3	7.0	7	16.3	5	11.6	23	53.5	2	4.7	2	4.7	43	100.0
	**8–11 yrs**	4	10.5	4	10.5	10	26.3	6	15.8	7	18.4	4	10.5	3	7.9	38	100.0
	**1–4 yrs**	0	0.0	1	2.3	1	2.3	10	23.3	17	39.5	11	25.6	3	7.0	43	100.0
**B**	**14–17 yrs**	0	0.0	0	0.0	12	30.8	11	28.2	14	35.9	2	5.1	0	0.0	39	100.0
	**8–11 yrs**	1	2.6	1	2.6	16	41.0	5	12.8	14	35.9	2	5.1	0	0.0	39	100.0
	**1–4 yrs**	3	8.8	1	2.9	1	2.9	1	2.9	25	73.5	3	8.8	0	0.0	34	100.0
**C**	**14–17 yrs**	0	0.0	1	7.7	5	38.5	1	7.7	5	38.5	0	0.0	1	7.7	13	100.0
	**8–11 yrs**	0	0.0	5	13.5	10	27.0	10	27.0	2	5.4	6	16.2	4	10.8	37	100.0
	**1–4 yrs**	0	0.0	6	17.7	3	8.8	12	35.3	8	23.5	2	5.9	3	8.8	34	100.0
**D**	**14–17 yrs**	0	0.0	4	10.5	14	36.8	9	23.7	5	13.2	0	0.0	6	15.8	38	100.0
	**8–11 yrs**	1	2.5	2	5.0	6	15.0	18	45.0	6	15.0	0	0.0	7	17.5	40	100.0
	**1–4 yrs**	0	0.0	5	10.6	7	14.9	25	53.2	10	21.3	0	0.0	0	0.0	47	100.0
**Subtotal** **(A–D)**	**14–17 yrs**	1	0.8	8	6.0	38	28.6	26	19.5	47	35.3	4	3.0	9	6.8	133	100.0
	**8–11 yrs**	6	3.9	12	7.8	42	27.3	39	25.3	29	18.8	12	7.8	14	9.1	154	100.0
	**1–4 yrs**	3	1.9	13	8.2	12	7.6	48	30.4	60	38.0	16	10.1	6	3.8	158	100.0
**E**	**14–17 yrs**	*-*	-	-	-	-	-	-	-	-	-	-	-	-	-	-	-
	**8–11 yrs**	0	0.0	0	0.0	0	0.0	24	63.2	2	5.3	2	5.3	10	26.3	38	100.0
	**1–4 yrs**	1	1.9	3	5.7	1	1.9	34	64.2	2	3.8	0	0.0	12	22.6	53	100.0
**F**	**14–17 yrs**	*-*	-	-	-	-	-	-	-	-	-	-	-	-	-	-	-
	**8–11 yrs**	0	0.0	4	10.0	1	2.5	27	67.5	0	0.0	8	20.0	0	0.0	40	100.0
	**1–4 yrs**	0	0.0	4	10.0	0	0.0	33	82.5	0	0.0	3	7.5	0	0.0	40	100.0
**Total** **(A–F)**	**8–11 yrs**	6	2.6	16	6.9	43	18.5	90	38.8	31	13.4	22	9.5	24	10.3	232	100.0
	**1–4 yrs**	4	1.6	20	8.0	13	5.2	115	45.8	62	24.7	19	7.6	18	7.2	251	100.0

Abbreviations. CAP: Child and Adolescent Psychiatrist. * More than one: more than one referrer among the options CAP, Hospital, School, Paediatrician and Family.

**Table 4 brainsci-12-00263-t004:** Frequency of subjects by neurodevelopmental concerns for referral, CAMH unit and age group.

CAMH	Age Group	ASD		Social		Language		ChBeh		Soc + Lang		Soc + ChBeh		Lang + ChBeh		Soc + Lang + ChBeh		Other		ASD + Other Symptoms		Total	
		n	%	n	%	n	%	n	%	n	%	n	%	n	%	n	%	n	%	n	%	n	%
**A**	**14–17 yrs**	1	2.3	12	27.3	5	11.4	6	13.6	7	15.9	5	11.4	2	4.6	2	4.6	4	9.1	0	0.0	44	100.0
	**8–11 yrs**	2	5.1	10	25.6	9	23.1	1	2.6	5	12.8	4	10.3	1	2.6	0	0.0	3	7.7	4	10.3	39	100.0
	**1–4 yrs**	6	14.0	11	25.6	4	9.3	0	0.0	7	16.3	1	2.3	2	4.7	0	0.0	2	4.7	10	23.3	43	100.0
**B**	**14–17 yrs**	9	23.1	5	12.8	2	5.1	1	2.6	11	28.2	2	5.1	0	0.0	5	12.8	1	2.6	3	7.7	39	100.0
	**8–11 yrs**	1	2.5	11	27.5	2	5.0	1	2.5	7	17.5	7	17.5	0	0.0	8	20.0	3	7.5	0	0.0	40	100.0
	**1–4 yrs**	10	24.4	5	12.2	3	7.3	0	0.0	20	48.8	1	2.4	0	0.0	0	0.0	0	0.0	2	4.9	41	100.0
**C**	**14–17 yrs**	0	0.0	0	0.0	5	38.5	1	7.7	3	23.1	0	0.0	0	0.0	1	7.7	2	15.4	1	7.7	13	100.0
	**8–11 yrs**	3	8.1	3	8.1	12	32.4	4	10.8	4	10.8	0	0.0	5	13.5	1	2.7	5	13.5	0	0.0	37	100.0
	**1–4 yrs**	4	11.8	2	5.9	6	17.6	8	23.5	0	0.0	0	0.0	8	23.5	3	8.8	1	2.9	2	5.9	34	100.0
**D**	**14–17 yrs**	5	31.3	0	0.0	7	43.8	1	6.3	0	0.0	0	0.0	0	0.0	0	0.0	3	18.8	0	0.0	16	100.0
	**8–11 yrs**	1	3.5	4	13.8	7	24.1	6	20.7	3	10.4	0	0.0	0	0.0	1	3.5	5	17.2	2	6.9	29	100.0
	**1–4 yrs**	7	14.9	4	8.5	19	40.4	1	2.1	4	8.5	0	0.0	3	6.4	1	2.1	8	17.0	0	0.0	47	100.0
**Subtotal (A–D)**	**14–17 yrs**	15	13.4	17	15.2	19	17.0	9	8.0	21	18.8	7	6.3	2	1.8	8	7.1	10	8.9	4	3.6	112	100.0
	**8–11 yrs**	7	4.8	28	19.3	30	20.7	12	8.3	19	13.1	11	7.6	6	4.1	10	6.9	16	11.0	6	4.1	145	100.0
	**1–4 yrs**	27	16.4	22	13.3	32	19.4	9	5.5	31	18.8	2	1.2	13	7.9	4	2.4	11	6.7	14	8.5	165	100.0
**E**	**14–17 yrs**	-	-	-	-	-	-	-	-	-	-	-	-	-	-	-	-	-	-	-	-	-	-
	**8–11 yrs**	20	50.0	5	12.5	3	7.5	2	5.0	5	12.5	3	7.5	1	2.5	0	0.0	0	0.0	1	2.5	40	100.0
	**1–4 yrs**	30	55.6	3	5.6	3	5.6	2	3.7	11	20.4	2	3.7	0	0.0	0	0.0	0	0.0	3	5.6	54	100.0
**F**	**14–17 yrs**	-	-	-	-	-	-	-	-	-	-	-	-	-	-	-	-	-	-	-	-	-	-
	**8–11 yrs**	3	7.5	6	15.0	3	7.5	0	0.0	11	27.5	7	17.5	1	2.5	1	2.5	1	2.5	7	17.5	40	100.0
	**1–4 yrs**	5	12.5	1	2.5	11	27.5	0	0.0	10	25.0	1	2.5	1	2.5	4	10.0	1	2.5	6	15.0	40	100.0
**Total** **(A–F)**	**8–11 yrs**	30	13.3	39	17.3	36	16.0	14	6.2	35	15.6	21	9.3	8	3.6	11	4.9	17	7.6	14	6.2	225	100.0
	**1–4 yrs**	62	23.9	26	10.0	46	17.8	11	4.2	52	20.1	5	1.9	14	5.4	8	3.1	12	4.6	23	8.9	259	100.0

ASD: Suspect autism spectrum disorder; Social: social impairment; Language: language impairment; ChBeh: challenging behaviours; Soc + Lang: social and language impairments; Soc + ChBeh: social impairment and challenging behaviours; Lang + ChBeh: language impairment and challenging behaviours; Soc + Lang + ChBeh: social impairment and language impairment and challenging behaviours; ASD + other symptoms: ASD and one or more other symptoms among social impairment, language impairment and challenging behaviours.

**Table 5 brainsci-12-00263-t005:** Age at diagnosis (median and range: min–max) of subjects by CAMH unit and age groups.

CAMH	Age Group	n	Median	min	max
**A**	**14–17 yrs**	44	4.03	1.34	16.04
	**8–11 yrs**	40	4.32	2.21	12.17
	**1–4 yrs**	43	2.97	1.44	5.31
**B**	**14–17 yrs**	38	4.75	2.33	13.90
	**8–11 yrs**	37	3.29	1.69	8.39
	**1–4 yrs**	41	2.82	1.18	4.99
**C**	**14–17 yrs**	12	5.40	2.49	8.05
	**8–11 yrs**	35	3.42	1.66	8.92
	**1–4 yrs**	32	2.38	1.28	4.53
**D**	**14–17 yrs**	38	3.47	1.38	13.97
	**8–11 yrs**	40	3.38	1.31	8.04
	**1–4 yrs**	45	2.81	1.24	4.46
**Subtotal** **(A–D)**	**14–17 yrs**	132	4.20	1.34	16.04
	**8–11 yrs**	152	3.72	1.31	12.17
	**1–4 yrs**	161	2.72	1.18	5.31
**E**	**14–17 yrs**	-	-	-	-
	**8–11 yrs**	40	5.02	1.97	9.84
	**1–4 yrs**	54	2.86	1.35	5.15
**F**	**14–17 yrs**	-	-	-	-
	**8–11 yrs**	40	5.87	3.41	10.63
	**1–4 yrs**	36	2.49	1.21	4.18
**Total**	**8–11 yrs**	232	4.30	1.31	12.17
**(A–F)**	**1–4 yrs**	251	2.74	1.18	5.31

**Table 6 brainsci-12-00263-t006:** Frequency of subjects by the composition/quality of the teams providing ASD diagnosis, CAMH unit and age group.

CAMH Unit	Age Group	Optimal		Suboptimal		Sufficient		Poor		Total	
		n	%	n	%	n	%	n	%	n	%
**A**	**14–17 yrs**	19	43.2	22	50.0	3	6.8	0	0.0	44	100.0
	**8–11 yrs**	34	85.0	6	15.0	0	0.0	0	0.0	40	100.0
	**1–4 yrs**	25	58.1	14	32.6	4	9.3	0	0.0	43	100.0
**B**	**14–17 yrs**	23	59.0	12	30.8	4	10.3	0	0.0	39	100.0
	**8–11 yrs**	18	45.0	15	37.5	7	17.5	0	0.0	40	100.0
	**1–4 yrs**	6	15.4	17	43.6	16	41.0	0	0.0	39	100.0
**C**	**14–17 yrs**	8	66.7	3	25.0	1	8.3	0	0.0	12	100.0
	**8–11 yrs**	30	100.0	0	0.0	0	0.0	0	0.0	30	100.0
	**1–4 yrs**	25	92.6	1	3.7	1	3.7	0	0.0	27	100.0
**D**	**14–17 yrs**	39	100.0	0	0.0	0	0.0	0	0.0	39	100.0
	**8–11 yrs**	40	100.0	0	0.0	0	0.0	0	0.0	40	100.0
	**1–4 yrs**	47	100.0	0	0.0	0	0.0	0	0.0	47	100.0
**Subtotal (A–D)**	**14–17 yrs**	89	66.4	37	27.6	8	6.0	0	0.0	134	100.0
	**8–11 yrs**	122	81.3	21	14.0	7	4.7	0	0.0	150	100.0
	**1–4 yrs**	103	66.0	32	20.5	21	13.5	0	0.0	156	100.0
**E**	**14–17 yrs**	-	-	-	-	-	-	-	-	-	-
	**8–11 yrs**	2	4.8	28	66.7	3	7.1	9	21.4	42	100.0
	**1–4 yrs**	0	0.0	46	83.6	0	0.0	9	16.4	55	100.0
**F**	**14–17 yrs**	-	-	-	-	-	-	-	-	-	-
	**8–11 yrs**	28	70.0	12	30.0	0	0.0	0	0.0	40	100.0
	**1–4 yrs**	40	100.0	0	0.0	0	0.0	0	0.0	40	100.0
**Total** **(A–F)**	**8–11 yrs**	152	65.5	61	26.3	10	4.3	9	3.9	232	100.0
	**1–4 yrs**	143	57.0	78	31.1	21	8.4	9	3.6	251	100.0

**Table 7 brainsci-12-00263-t007:** Frequency of subjects assessed by the use of standardized tools for ASD diagnosis, cognitive function, adaptive skills and definition of psychoeducative profile by CAMH units and age groups.

CAMH Unit	Age Group	Subjects	ASD Diagnosis ^a^		Cognitive ^b^		Adaptive Skills ^c^		Psychoeduc. Profile ^d^	
		n	n	%	n	%	n	%	n	%
**A**	**14–17 yrs**	41	19	46.3	36	87.8	7	17.1	3	7.3
	**8–11 yrs**	38	30	79.0	33	86.8	2	5.3	0	0.0
	**1–4 yrs**	42	38	90.5	40	95.2	3	7.1	0	0.0
**B**	**14–17 yrs**	28	19	67.9	13	46.4	4	14.3	1	3.6
	**8–11 yrs**	34	17	50.0	11	32.4	11	32.4	16	47.1
	**1–4 yrs**	36	31	86.1	1	2.8	26	72.2	34	94.4
**C**	**14–17 yrs**	9	5	55.6	6	66.7	2	22.2	0	0.0
	**8–11 yrs**	31	16	51.6	27	87.1	6	19.4	1	3.2
	**1–4 yrs**	17	17	100.0	8	47.1	3	17.7	0	0.0
**D**	**14–17 yrs**	35	21	60.0	30	85.7	23	65.7	0	0.0
	**8–11 yrs**	40	33	82.5	32	80.0	9	22.5	0	0.0
	**1–4 yrs**	46	45	97.8	18	39.1	0	0.0	0	0.0
**Subtotal** **(A–D)**	**14–17 yrs**	113	64	56.6	85	75.2	36	31.9	4	3.5
	**8–11 yrs**	143	96	67.1	103	72.0	28	19.6	17	11.9
	**1–4 yrs**	141	131	92.9	67	47.5	32	22.7	34	24.1
**E**	**14–17 yrs**	-	-	-	-	-	-	-	-	-
	**8–11 yrs**	42	41	97.6	22	52.4	0	0.0	11	26.2
	**1–4 yrs**	55	55	100.0	43	78.2	3	5.5	11	20.0
**F**	**14–17 yrs**	-	-	-	-	-	-	-	-	-
	**8–11 yrs**	40	40	100.0	38	95.0	3	7.5	0	0.0
	**1–4 yrs**	39	39	100.0	24	61.5	0	0.0	8	20.5
**Total**	**8–11 yrs**	225	177	78.7	163	72.4	31	13.8	28	12.4
**(A–F)**	**1–4 yrs**	235	225	95.7	134	57.0	35	14.9	53	22.55

^a^ At least one standardized tool among ADOS-G, ADOS-2, CARS and CASD. ^b^ At least one standardized tool among GRIFFITHS, BAYLEY, LEITER, WPPSI-III and WISC-III. ^c^ VABS. ^d^ At least one standardized tool among PEP, PEP-3 and ESDM.

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
