# Peer review of "Management of Autism Spectrum Disorder in Italian Units of Child and Adolescent Mental Health: Diagnostic and Referral Pathways"

_brainsci, 2022, doi:10.3390/brainsci12020263_

Round 1

Reviewer 1 Report

Brief summary:

The present study provides detailed information on clinical management for ASD referral and diagnosis that are mandatory for child and adolescent mental health management. The aim of the study is to estimate if the international recommendations are applied. They included subjects born in three different calendar year periods, to verify if management processes have changed over time. Data collected on 634 children and 165 adolescents with ASD showed a high variability among CAMH units and suggest some improvements that CAMH units should implement, namely establishing a multidisciplinary team for diagnosis; improving the assessment of physical and mental conditions by the use of standardized tools and implement a specific assessment for challenging behaviors.

It’s an interesting study on Italian CAMH’s practices with ASD patients. The manuscript is clear and well-structured.

As they support their investigation on international recommendations and IAPP plan, it would have been interesting to define international (not only the NICE) and national standards in the introduction. Nothing is said about ADI-R within the evaluation.

In addition, in clinical practice, the recommendations are not the same depending of the age of evaluation. In addition, the additional examens such as MRI, genetics, .. are questioning in general clinical practice in ASD without any research objectives.

The main limitation of the present study was the low number of CAMH units included.

The references are adapted but might be increased by other gold standard than the NICE.

  • Is the manuscript scientifically sound and is the experimental design appropriate to test the hypothesis? yes
  • Are the manuscript’s results reproducible based on the details given in the methods section? Not really as it depends of the Italian CAMH organization
  • Are the figures/tables/images/schemes appropriate? Yes Do they properly show the data? Yes Are they easy to interpret and understand? Yes  Are the data interpreted appropriately and consistently throughout the manuscript? Yes Please include details regarding the statistical analysis or data acquired from specific databases.
  • Are the conclusions consistent with the evidence and arguments presented? Yes
  • Please evaluate the ethics statements and data availability statements to ensure they are adequate. Nothing is precise about ethical consideration in data collection ?

Author Response

Reviewer 1.

Brief summary:

The present study provides detailed information on clinical management for ASD referral and diagnosis that are mandatory for child and adolescent mental health management. The aim of the study is to estimate if the international recommendations are applied. They included subjects born in three different calendar year periods, to verify if management processes have changed over time. Data collected on 634 children and 165 adolescents with ASD showed a high variability among CAMH units and suggest some improvements that CAMH units should implement, namely establishing a multidisciplinary team for diagnosis; improving the assessment of physical and mental conditions by the use of standardized tools and implement a specific assessment for challenging behaviours.

It’s an interesting study on Italian CAMH’s practices with ASD patients. The manuscript is clear and well-structured.

As they support their investigation on international recommendations and IAPP plan, it would have been interesting to define international (not only the NICE) and national standards in the introduction.

We are aware of the several further international standards beyond those of UK (e.g. National Clearinghouse on Autism Evidence; New Zealand Autism Spectrum Disorder Guideline 2016, 2019; the National guideline for the assessment and diagnosis of autism spectrum disorders in Australia, 2018). However, we specifically cited the SIGN recommendations because the IAPP was based mainly on the current Italian guideline for the management of children and adolescents with autism (SNLG 2011), which was developed as an upgrading of the SIGN guideline n. 98, 2007. We added ae sentence clarifying this point (lines 72-73)

As for NICE Quality Standards QS15, we considered these standards as they include both the main management milestones and the definition of qualitative and quantitative indicators for the assessment of quality of ASD management. It appeared interesting as benchmark for discussing our data. However, we added your observation as future perspective at the end of the Limitation chapter:

“Finally, It would be of particular interest to examine our data also taking into account different standards of autism that are developed at international levels (Ministries of Health and Education, 2016; Whitehouse et al 2018; Brian et al 2019)”

Thank you to point out the lack of definition of standards in the Introduction. We added a sentence that in a very synthetic way resumed the main recommendations  that are considered  a requisite for ASD management and that are shared at Italian and international levels. Please see lines 73-79: “In particular, these documents stated the fundamental principles for ASD management: i) capillary early diagnostic processes; ii) comprehensive, easily accessible and widespread health services network throughout the territory; iii) multi-professionalism and interdisciplinary synergy for the intervention that services must be able to offer; iv) strong integration of the health, social, school and educational dimensions; v) continuity of taking charge for the entire life of the person, with the progressive adaptation of interventions and the organization of living spaces.”

In addition, in clinical practice, the recommendations are not the same depending of the age of evaluation.

We agree with the observation of the referee. However,  in our opinion this is particularly relevant for how CAMH units manage the intervention, a piece of information that is not analysed in the present study whose focus is in the pathways of referral and diagnosis. The diagnostic process, that usually should take place early in life, must include the evaluation of ASD symptoms, and the functional assessment (e.g. cognitive and adaptive skills). This must be done whatever the age of the child at the moment of diagnosis. Child’s age only affects the choice of the tools used for such assessments.

Regarding the laboratory tests, age at evaluation is crucial for the choice of the test. However, it should be taken into account that the analysis of each group of age is aimed at estimating the clinic behaviour in a particular historic time rather than in a particular age. Each age group includes children born in a specific calendar year interval, allowing to observe which kind of clinical practices were implemented in that historical period. Indeed, we stressed in the result and discussion sections that some tests, more frequently used in the 1-4 years group i.e. in the group of children more recently assessed, could be driven by research objectives (see Discussion, lines 450-460).

In addition, the additional examens such as MRI, genetics,.. are questioning in general clinical practice in ASD without any research objectives.

We completely agree with the referee. We extensively argument this specific question in the Discussion (lines 444-470).

Nothing is said about ADI-R within the evaluation.

We are very grateful to the reviewer for noting the missing information about ADI-R. We added a paragraph in the Methods (Measures) to better introduce the analysis of the use of standardised tools for diagnostic and functional evaluation (please see lines 178-193). More in particular “Standardized tools used during the diagnostic process were categorized as tools for: i) diagnosis (ADOS-G, ADOS2, CARS, and CASD); ii) cognitive functions (GRIF-FITHS, BAYLEY, LEITER, WPPSI-III, WISC-III); iii) adaptive skills (VABS); iv) definition of the psychoeducational profile (PEP, PEP-3, ESDM). For each category, we computed the frequency of subjects undergoing the evaluation by at least one of the tools listed in the category. Autism Diagnostic Interview-Revised (ADI-R) was not includ-ed in the diagnosis category as this tool resulted to be very rarely used in the participating CAMH units (n=20) and never alone but always in combination with ADOS or CARS.”

The main limitation of the present study was the low number of CAMH units included.

We agree with the reviewer, indeed we specified this point in the Discussion (Paragraph Limitations ). It should be considered that this represents a pilot study aimed to build a systematic protocol to survey CAMH units activities dedicated to ASD referral and diagnosis in Italy. Thus, the low number of CAMH units limits the possibility for a conclusive assessment of the clinical behaviour in Italy for referral and diagnosis, or to identify specific predictive factors of specific clinical behaviour. However, the sample under study is sufficient to support the use of a systematic protocol to describe the variation between CAMH units and change over time, and this represent a tool to improve accountability of the clinical behaviour for the processes for ASD referral and diagnosis.

The references are adapted but might be increased by other gold standard than the NICE

We added further references about published international standards (please see references in yellow)

  • Is the manuscript scientifically sound and is the experimental design appropriate to test the hypothesis? yes
  • Are the manuscript’s results reproducible based on the details given in the methods section? Not really as it depends of the Italian CAMH organization
  • Are the figures/tables/images/schemes appropriate? Yes
  • Do they properly show the data? Yes
  • Are they easy to interpret and understand? Yes  
  • Are the data interpreted appropriately and consistently throughout the manuscript? Yes 
  • Please include details regarding the statistical analysis or data acquired from specific databases.
  • Are the conclusions consistent with the evidence and arguments presented? Yes

  • Please evaluate the ethics statements and data availability statements to ensure they are adequate. Nothing is precise about ethical consideration in data collection ?

The following statement was added as requested by editorial standards (lines 555-558)

Institutional Review Board Statement: The study was conducted according to the guidelines of the Declaration of Helsinki. The procedure was approved by the Istituto Superiore di Sanità ethics committee (PRE BIO CE n. 17162/2018). Informed consent was obtained from all subjects involved in the study.

Reviewer 2 Report

The paper addresses important topic on the referral and diagnosis of autism spectrum disorder (ASD). It provides detailed information on the mandatory clinical management and diagnosis processes at the Italian child and adolescent mental health (CAMH) units and verifies if they adhere to the international recommendations for ASD referral and diagnosis. The paper is interesting. However there are several major difficulties with the ways the findings are reported and written.  I am presenting them numerically below so that it is easier to respond to them. Please see my comments below for the consideration:

Major compulsory revisions:

  1. It is hard to read this work. Many sentences are unclear. Thus a thorough revision of English language in this paper is needed. MDPI services are suggested.
  2. Line 19: the abbreviation “ASD” appears the first time in the text thus it should be additionally explained, written as (autism spectrum disorder).
  3. Line 99: the numerical subsections are not needed e.g. 2.1. etc. The subtitles only written in cursive as it is now e.g. Participants and Procedure etc. will be sufficient and make this text easier to read.
  4. Line 143: Table 2 and supplemental material: Information collected via structured Electronic Collection Protocol (ECP): I suggest to delete this table as it is unclear. Instead, the authors should describe ECP components from this table in more details in the text to non-Italian speaking readers especially as ECP has been presented in the supplemental material only in Italian language.
  5. Line 169: Table 3. Distribution of subjects by CAMH unit and age group, and percentage of male subjects. Please explain in the text (and definitely in the limitations of this study) why there were no female subjects presented and only males. Address this major limitation in the text and create future directions. The line 309 brings even more confusion on females representation in this study.
  6. Line 177: explain why the authors could not present the data that they are referring to.
  7. Line 230: supplemental Table S5 should be deleted instead more details should be written in the text about the criteria for each multidisciplinary team: optimal, sub-optimal, good, sufficient, poor.
  8. Line 248: all the abbreviations for the assessment instruments appear here for the first time thus should be also written as e.g. ADOS-2 (Autism Diagnostic Observation Schedule, Second Edition) etc.
  9. The authors should delete the redundant: “data are shown in supplementary materials” in the whole text wherever they refer to this material. Example: line 275 or 388 etc. Only (Table S7) is sufficient.
  10. Lines 296-304 should be moved to the Introduction paragraph.
  11. Line 309 (Discussion): “the male-to-female ratio observed in our sample ranged between 4:1 and 5:1”. This is the first time authors mention about female subjects. Additionally the Table 3. Distribution of subjects by CAMH unit and age group, and percentage of male subjects presents only males. It is confusing and authors needs to clarify it.
  12. The discussion section presents flows. The authors should move some information from this section to the introduction section while in the discussion they should comment on their results.
  13. Line 397: authors need to write exactly which ICD and/or DSM criteria are recommended for this process.

Minor essential revisions:

Please correct the abbreviations, grammar and spelling in the paper. Only a few examples are provided below:

  1. Line 32 (Keywords) or line 207, 209, 284 or 417 etc.: capital and small letters should be corrected, grammar and spelling as well. Again, MDPI or other services are suggested to improve English language in this paper.

Author Response

Reviewer 2

The paper addresses important topic on the referral and diagnosis of autism spectrum disorder (ASD). It provides detailed information on the mandatory clinical management and diagnosis processes at the Italian child and adolescent mental health (CAMH) units and verifies if they adhere to the international recommendations for ASD referral and diagnosis. The paper is interesting. However, there are several major difficulties with the ways the findings are reported and written. I am presenting them numerically below so that it is easier to respond to them. Please see my comments below for the consideration:

Major compulsory revisions:

  1. It is hard to read this work. Many sentences are unclear. Thus a thorough revision of English language in this paper is needed. MDPI services are suggested.

We checked English editing as suggested by the reviewer. Moreover, as requested by the Editorial office of the MDPI, in order to accelerate publication time, a more extensive English editing will be performed after the revision process and before publication, with the support of the editorial office of the MDPI.

  1. Line 19: the abbreviation “ASD” appears the first time in the text thus it should be additionally explained, written as (autism spectrum disorder).

We have now explained the abbreviation ASD at Line 19

  1. Line 99: the numerical subsections are not needed e.g.1.etc. The subtitles only written in cursive as it is now e.g. Participants and Procedure etc. will be sufficient and make this text easier to read.

We removed numerical subsections as suggested by the reviewer for each Chapter of the manuscript.

  1. Line 143: Table 2and supplemental material: Information collected via structured Electronic Collection Protocol (ECP): I suggest to delete this table as it is unclear. Instead, the authors should describe ECP components from this table in more details in the text to non-Italian speaking readers especially as ECP has been presented in the supplemental material only in Italian language.

As suggested by the reviewer we have cancelled table 2 and included more details on ECP variables in the text (please see Paragraph “Measures” in the Methods section)

  1. Line 169: Table 3. Distribution of subjects by CAMH unit and age group, and percentage of male subjects. Please explain in the text (and definitely in the limitations of this study) why there were no female subjects presented and only males. Address this major limitation in the text and create future directions. The line 309 brings even more confusion on females representation in this study.

We are sorry, the title of the Table 3 was misleading, we collected data on both males and females. We have now changed the title of Table 3 and added more columns to describe the sex distribution in each age group and CAMH. We also discussed on sex ration data in the Discussion (lines 373-377).

  1. Line 177: explain why the authors could not present the data that they are referring to.

We have now added the percentages of children and adolescents having challenging behaviours in the three age groups. Lines 239-244.

  1. Line 230: supplemental Table S5 should be deleted instead more details should be written in the text about the criteria for each multidisciplinary team: optimal, sub-optimal, good, sufficient, poor.

We have deleted Table S5 and added a sentence in the Method section, explaining the criteria that we used to classify the multidisciplinary team. Lines 195-210.

  1. Line 248: all the abbreviations for the assessment instruments appear here for the first time thus should be also written as e.g. ADOS-2 (Autism Diagnostic Observation Schedule, Second Edition) etc.

We explained the abbreviations as indicated by the reviewer in the Method section (Measures, lines 178-187)

  1. The authors should delete the redundant: “data are shown in supplementary materials” in the whole text wherever they refer to this material. Example: line 275 or 388 etc. Only (Table S7) is sufficient.

We changed the text according to the reviewer suggestion.

  1. Lines 296-304 should be moved to the Introduction paragraph.

Following the reviewer indication, we moved the sentence “In Italy, CAMH units have a capillary distribution, each one serving a population of about 100,000 residents with a percentage of residents under 18 years of age which strongly depends on time and geographical area, with a mean value of about 16% in 2017” from Discussion to the Introduction.

  1. Line 309 (Discussion): “the male-to-female ratio observed in our sample ranged between 4:1 and 5:1”. This is the first time authors mention about female subjects. Additionally, the Table 3. Distribution of subjects by CAMH unit and age group, and percentage of male subjects presents only males. It is confusing and authors needs to clarify it.

We thank the reviewer to have noted the misleading reporting of the sample characterisation by sex. See answer to point 5.

  1. The discussion section presents flows. The authors should move some information from this section to the introduction section while in the discussion they should comment on their results.

We thank the reviewer for the advice. To improve the reading of the discussion and better focus on our results, we rephrased some sentences and moved the following paragraphs from the Discussion to the Introduction section, adapting them to the target text:

“In Italy, CAMH units have a capillary distribution, each one serving a population of about 100,000 residents with a percentage of residents under 18 years of age which strongly depends on time and geographical area, with a mean value of about 16% in 2017” and modified according to the text”.

“Several scholars and guidelines (Zwaigenbaum& Maguire 2019; Hyman et al. 2020) have underlined the importance of early identification and referral for diagnostic evaluation and intervention services. Indeed, early diagnosis - paired with a timely intervention - appears to have a positive impact on life trajectory, even if its efficacy in reducing autism severity is low (Vivanti et al. 2017, Nahmias et al. 2019). Hence, many countries, including Italy, are putting into place different actions to improve neurodevelopmental surveillance and increase knowledge and awareness about ASD symptoms and their early manifestation, mainly through training programs for health professionals and teachers, and by promoting the use of standardised checklists”.

“The need for a multidisciplinary team to ensure an accurate diagnosis of neurodevelopmental disorders including ASD is well-recognized. The recent SIGN evidence-based guideline (SIGN 2016) stated that “diagnostic assessment, alongside a profile of the individual’s strengths and weaknesses, carried out by a multidisciplinary team which has the skills and experience to undertake the assessments, should be considered as the optimum approach for individuals suspected of having ASD””.

  1. Line 397: authors need to write exactly which ICD and/or DSM criteria are recommended for this process.

Line 407 – We rephrased the sentence to address the reviewer’s comment: “The diagnosis of ASD is performed by differentiating between two or more conditions that share similar signs or symptoms. ICD 11 and/or DSM 5 are the more recent available manuals to conduct an accurate differential diagnosis process.

Minor essential revisions:

Please correct the abbreviations, grammar and spelling in the paper. Only a few examples are provided below:

  1. Line 32 (Keywords) or line 207, 209, 284 or 417 etc.: capital and small letters should be corrected, grammar and spelling as well. Again, MDPI or other services are suggested to improve English language in this paper.

We amended the text according to the reviewer's suggestions.

Round 2

Reviewer 2 Report

The authors did a lot of work making revisions. The changes that the authors made helped to strengthen the paper while providing a clearer picture to the reader.

Please see my comments below for the minor corrections:

  • Lines 165, 171, 234, 436 etc.: correct ICD10 to ICD-10; ICD 11 to ICD-11 and DSM 5 to DSM-5.
  • Lines 179,180, 313, 326 etc.: Autism Diagnostic Observation Schedule-Generic2nd Edition (ADOS2) - correct to: Autism Diagnostic Observation Schedule, Second Edition (ADOS-2).
  • Line 436: I suggest to rephrase this sentence for example: ICD-11 and /or DSM-5 define and classify mental disorders in order to improve diagnoses, treatment, and research (…).
  • Line 536: (…) different standards of autism (…). I suggest to rephrase this unfortune wording to e.g.: different standards of diagnostic assessment for ASD etc.

Author Response

Thank you very much for the very accurate critical review of the manuscript that has allowed its enhancement We addressed minor revision suggested, namely:

  • Lines 165, 171, 234, 436 etc.: we changed ICD10 to ICD-10; ICD 11 to ICD-11 and DSM 5 to DSM-5.
  • Lines 179,180, 313, 326 etc.: we changed Autism Diagnostic Observation Schedule-Generic2nd Edition (ADOS2) – to Autism Diagnostic Observation Schedule, Second Edition (ADOS-2).
  • Line 436: we rephrased the sentence as suggested: “ICD-11 and /or DSM-5 define and classify mental disorders in order to improve diagnoses, treatment, and research (…).
  • Line 536: (…) we apologise for the inappropriate wording. We rephrased the sentence following reviewer suggestion: “different standards of diagnostic assessment for ASD (…).